# Cataract Surgery after Radial Keratotomy with Non-Diffractive Extended Depth of Focus Lens Implantation

**DOI:** 10.3390/medicina58050689

**Published:** 2022-05-23

**Authors:** Anna Dołowiec-Kwapisz, Marta Misiuk-Hojło, Halina Piotrowska

**Affiliations:** 1Department of Ophthalmology, Hospital in Zgorzelec, 59-900 Zgorzelec, Poland; okulistyka@spzoz.zgorzelec.pl; 2Department of Ophthalmology, Wrocław Medical University, 50-556 Wrocław, Poland; marta.misiuk-hojlo@umw.edu.pl

**Keywords:** radial keratotomy, cataract, EDOF, phacoemulsifacation, cataract surgery

## Abstract

Radial keratotomy was a popular surgical procedure used to treat myopia. Patients who underwent radial keratotomy several years ago, are currently reporting to the ophthalmologist due to worsening of vision associated with age-related cataracts. In this case report we present a case of a 60-year-old woman who underwent radial keratotomy with 16 incisions in the right eye and 12 incisions in the left eye. The patient reported to an ophthalmologist due to a deterioration of vision caused by a cataract. We described, in detail, the difficulties encountered during the diagnostic procedures, differences in the calculation of intraocular lens, and intraoperative difficulties as compared to patients who had not undergone radial keratotomy. We also present the obtained postoperative results.

## 1. Introduction

In the 1980s and 1990s, radial keratotomy (RK) was a popular surgical procedure used to treat myopia. The aim of RK is to flatten the central curvature of the cornea. This effect is achieved by making radial incisions in the cornea (usually 8–16), the depth of which reach 90% of the corneal thickness. In this manner, the procedure changes the curvature of the anterior and posterior surfaces of the cornea [1]. However, more and more patients, who had undergone RK in the past, report to the ophthalmologist due to worsening of vision caused by age-related cataracts.

Patients who underwent RK are a unique group of patients. They pose a special challenge for cataract surgeons at every stage of treatment, starting from the selection of the proper type of intraocular lens (IOL), calculation of IOL power, choosing the best place for the corneal incision, and ending with months of postoperative follow-up during which refraction may change constantly [2]. The expectations of patients, who want to reduce spectacle dependence after cataract surgery, are an additional challenge which may not always be met.

## 2. Case Report

A 60-year-old patient was admitted to the Department of Ophthalmology of the Voivodeship Hospital in Zgorzelec due to a deterioration of vision in both eyes seen over several months. In 1989 she underwent bilateral myopic RK (no documentation was available). She reported that before the procedure she had used corrective glasses with a power of −4,5 Dsph. After RK she did not require any spectacle correction for the next 15 years. In 2004, the patient started wearing progressive glasses again. On the day of admission, before cataract surgery, her refraction was: OD (right eye) +4.25/−0.75 ax 159, OS (left eye) +3.5/−0.5 ax 5; and visual acuity OD UCDVA (uncorrected distance visual acuity) at 4 m 1.0 logMAR, BCDVA (best corrected distance visual acuity) 0.2 logMAR (+4.0 Dsph), UCIVA (uncorrected intermediate visual acuity) at 80 cm 1.1 logMAR, BCIVA (best corrected intermediate visual acuity) 0.1 logMAR, UCNVA (uncorrected near visual acuity) at 40 cm 1.1 logMAR, BCNVA (best corrected near visual acuity) 0.3 logMAR; OS: UCDVA 1.0 logMAR, BCDVA 0.2 (+3.25 Dsph), UCIVA 1.1 logMAR, BCIVA 0.1 logMAR, UCNVA 1.0 logMAR, BCNVA 0.2 logMAR. The ophthalmological examination revealed incisions made during RK in the anterior segment—in OD 16 and OS 12 (Figure 1) Anterior and posterior segments of the eye were normal. The patient was informed in detail about the difficulties in calculating IOL power after the RK procedure and that corneal regeneration after cataract surgery takes more time. The possibility of postoperative refractive error was explained to the patient. The patient was qualified for cataract surgery, after conducting comprehensive ophthalmological examinations such as: biometry on the Argos SS-OCT biometer (Movu, Inc., Komaki, Japan) and IOL Master 500 (Carl Zeiss Meditec AG, Jena, Germany) (Figure 2), corneal tomography on scanning system Oculazer™ WaveLight^®^ II (Alcon Laboratories, Inc., Fort Worth, TX, USA) (Figure 3), optical coherence tomography (OCT) (OCT III, Carl Zeiss Meditec AG, Jena, Germany) of the anterior and posterior segment. Implantation of a non-diffractive lens with an extended depth of focus (EDOF), Acrysof IQ Vivity (Alcon Laboratories, Inc., Fort Worth, TX, USA) was chosen.

Only the OS corneal curvature was measured using the Argos optical biometer. We were unsuccessful in measuring the curvature in OD. Both eyes were examined without any problems on the IOL Master 500 biometer. To calculate the implant power on the Argos optical biometer, we used results of K1 and K2 from IOL Master (Figure 4). Considering the presence of the irregular and asymmetrical astigmatism of low value, seen in the topography, the implantation of a toric lens was rejected. 

The results of IOL power, calculated using the Barret’s True K formula was obtained with the use of an online calculator: the IOL Calculator for Eyes with Prior RK, which was developed by the American Society of Cataract and Refractive Surgery (ASCRS) (Figure 5). The Argos optical biometer was used. Both measurements were similar. Due to the lack of documentation from the RK surgery and a low amount of data which could be entered into the calculator, we chose the IOL power by averaging the measurement from IOL Master, the Argos optical biometer and the ASCRS calculator.

In September 2021 cataract surgery of OD, was performed. Implantation of IOL Acrysof IQ Vivity with a power of 24.0 D was used. Two weeks after the surgery, during a follow up visit, we obtained the following refractions: OD: +1.75/−1.75 axis 144, UCDVA 0.2 log mar, BCDVA 0.0 logMAR (+1.25/−1.0 ax 145), UCIVA 0.4 logMAR, BCIVA 0.14 logMAR, UCNVA 0.6 logMAR, BCNVA 0.1 logMAR (Table 1). After one month, during the control visit, the refraction was OD: +0.25/−1.25 ax 142, and the patient gained visual acuity: UCDVA 0.1 logMAR, BCDVA 0.0 logMAR (−0.75 ax 145)), UCIVA 0.3 logMAR, BCIVA 0.1 logMAR, UCNVA 0.6 logMAR, BCNVA 0.1 logMAR. In November 2021 cataract surgery of OS was performed using implantation of IOL Acrysof IQ Vivity with a power of 23,5D. Both procedures were performed by the same surgeon who used an Infinity phacoemulsifier from Alcon. No complications were seen after both surgeries. Two weeks after the second procedure we received the following refraction: OS −0.0/−0.5 ax 73, and the patient’s visual acuity was OS UCDVA 0.0 logMAR, BCDVA −0.1 log MAR (−0.25 Dsph), UCIVA 0.5 logMAR, BCIVA 0.1 log MAR, UCNVA 0.5 logMAR BCNVA 0.1 logMAR. Another control visit was conducted 6 weeks after the surgery of the second eye. Refraction and visual acuity are displayed in Table 1.

After both procedures the patient did not report any dysphotopsias and was pleased with the effects of the operation. Currently the patient does not require spectacles for distance or intermediate distance. She only uses power lenses for small print or in poor quality light.

The last control visit was conducted 6 months after the last surgery in order to evaluate the possible refractive changes connected to past RK. Table 2 presents the results of the control visit.

## 3. Discussion

The procedure of cataract surgery in patients who underwent RK is challenging for cataract surgeons due to difficulties in preoperative measurements of IOL as well as high expectations of patients regarding good vision after surgery. Earlier studies [3,4] showed that the refraction results after cataract surgery are difficult to predict in this group of patients.

### 3.1. Choice of the IOL

Our patient wanted to be independent of spectacles, however she was afraid of intensified visual disturbances such as halo and glare. After the RK procedure she reported dysphotopsia under meso and scotopic conditions—glare and halo. Difficulties in determining the curvature of the cornea and performing the calculation of the appropriate IOL power were caused by corneal irregularities after RK. It was difficult to predict the residual refractive error after the planned cataract surgery. In the case of multifocal lenses, which minimize the need to wear glasses, it is crucial to carefully select IOL power and to correct astigmatism, in order to achieve good visual acuity. Moreover, multifocal lenses may exert side effects like halo and glare, which may be enhanced by corneal aberrations in post-RK patients. The patient was advised not to choose multifocal lenses due to these reasons. Instead, the patient was offered an EDOF lens, Vivity.

The IOL provides an extended range of vision from a distance with excellent intermediate and functional near vision. It is based on non-diffractive X-wave technology, which modifies the wave front and produces one elongated focus without splitting light. Thanks to these properties, the lens reduces the risk of dysphotopsia. It does not lessen the contrast sensitivity and is less sensitive to decentration than multifocal lenses. This lens is built from Acrysof, a hydrophobic material, and contains UV and blue light filters. It has −1.5 D defocus and negative asphericity of the anterior surface (−0.2 μm) which is particularly important in patients with positive corneal aberrations. The difference between the results of the autorefractometer and the actual postoperative refractive error is typical for Vivity lenses. Hence, it is recommended to calculate the refraction manually with maximum plus technique [5]. The EDOF lenses appear to be an opportunity for post-RK patients who want some independence from spectacles. The extended depth of focus, such as that seen in the Acrysof IQ Vivity non-diffractive lens, can “forgive” the imperfection of IOL power selection caused by the difficulty in calculating IOL power in post-RK patients. Thanks to the elongated focal point and the resulting broadened defocus curve, patients can achieve acceptable UCVA (uncorrected visual aquity) levels over a larger residual refractive error width. This is a very important consideration in patients after RK surgery due to postoperative residual refractive error fluctuations.

### 3.2. Choice of IOL Power Calculation

Corneal incisions performed during RK procedures may contribute to greater measurement errors when standard methods are employed, especially regarding parameters necessary for the correct selection of the IOL (thickness and breaking power of the central part of the cornea). False measurement results of refractive power of the cornea come from the differences between the central flattened area of cornea after RK (3 mm) and the area which is measured by keratometer (up to 4 mm) [3]. Measurement errors may lead to incorrect calculations of effective lens position (ELP) and IOL power which may, in turn, result in postoperative hyperopia [6]. Newer third and fourth generation calculating formulas enable better ELP estimation [7,8]. Turnbull et al. recommend the use of the following formulas in order to calculate the IOL power—depending on available data before and after RK,

- in the case of available medical history before and after a RK procedure:

Barrett True K [History]

Barrett True K [Partial History]

- in the case of the missing information in medical history before and after RK procedure:

Barrett True K [No History]

Standard Haigis formula.

In the study conducted by this group of authors, the method of True K and the standard Haigis formula were able to give much better results during the postoperative period than the DK-Holladay-IOLM, Potvin-Hill or Haigis methods (with shift of −0.5D) [9].

Thanks to the IOL ASCRS calculator, which is available online, one can calculate the lens power after RK and LASIK/PRK procedures (also after both hyperopia correction and myopia correction). They are easy to use and widely available. The introduction of data into the online calculator is used to calculate the IOL power by using seven different calculating formulas. The best solution seems to be the use of the averaged result from all available formulas [10].

It should be kept in mind that the RK procedure not only changes the corneal curvature but may also create alternating flat and convex zones on the corneal surface. This makes it difficult to determine the flat and steep meridian of the cornea [11]. Results of following keratometries (no matter what kind of apparatus had been used) are not repetitive and may differ from each other. This may affect the result of IOL power calculation and may result in refractive errors.

### 3.3. Incision Type

Cataract phacoemulsification increases the risk of RK incisions dehiscence or rupture [12,13]. Cases of ruptures or dehiscence of scars seen during surgical procedures such as retinal detachment surgery, phacoemulsification of cataracts or corneal transplant have been described in literaturę [14]. Even many years after RK surgery, the cornea does not return to its original integrity, and the scar tissue may contain corneal epithelial cells [15]. That is why it is crucial to properly plan the location of the cuts and maintain special caution during cataract phacoemulsification.

In this case the main incisions were made more circumferentially than in standard cataract surgery in order to minimize above the risks. Lateral incisions were made between two neighbouring scars after RK (Figure 6).

Meduri et al. showed that it was valid to place an incision within the cornea only in cases when there was enough room to cut between adjacent RK scars without disturbing their continuity. Disturbing the continuity of the RK scars may lead to scar dehiscence and leakage of aqueous humor. In the case when the location of an incision after RK start to open or leak, it should be sealed with a 10-0 nylon suture and the sutures should not be removed for 2 weeks [16]. The cataract removal procedure should be as short as possible due to reduced corneal stability after RK. The operator should restrict the movements of the phacoemulsifier head within the anterior chamber in order to prevent postoperative astigmatism [10].

### 3.4. Refractive Outcomes of Phacoemulsification in Post-RK Eyes

There are only a few publications on EDOF lens implantation in patients after radial keratotomy. This is primarily due to surgeons’ fear of implanting optically advanced intraocular lenses in patients who present corneal ‘multifocality’.

A study by Bartman et al. described the postoperative outcomes of 12 patients with a history of radial keratotomy (24 eyes) after implantation of an EDOF, Tecnis Symfony lens. They described an improvement in UCVA from 20/73 (Snellen equivalent) to 20/33 after 6 months and an average manifest SE that improved from +1.68D (preoperative) to −0.18 D (6 months after surgery). A high degree of patient satisfaction after the surgery was obtained [2].

Agarwal et al. described 2 cases of unilateral implantation of the IC-8 IOL in patients after bilateral radial keratotomy and 1 case after bilateral radial keratotomy and astigmatic keratotomy (AK) achieving good UCDVA, UCIVA and UCNVA in most cases [17].

## 4. Conclusions

Cataract surgery in patients who underwent RK is challenging for cataract surgeons due to difficulties encountered while choosing IOL power, planning the location of corneal incisions and a prolonged corneal regeneration after the procedure. It is hard to meet patients’ expectations when it comes to achieving spectacle independence after the procedure. The non-diffractive EDOF lenses give a chance to achieve satisfactory postoperative effects, while avoiding the typical side effects seen after multifocal lenses.

## Figures and Tables

**Figure 1 medicina-58-00689-f001:**
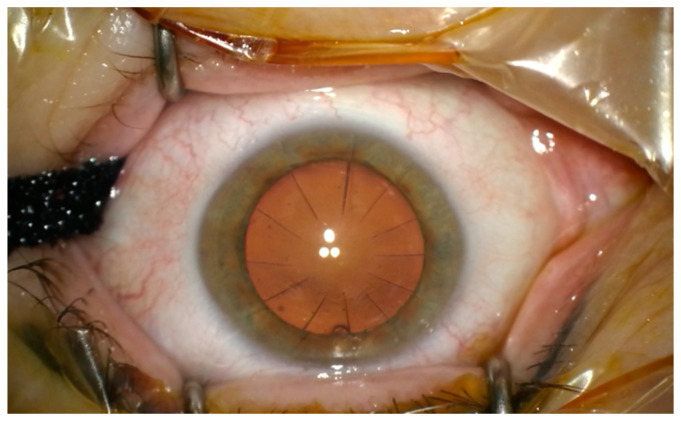
The picture shows 12 incisions after radial keratotomy in the left eye.

**Figure 2 medicina-58-00689-f002:**
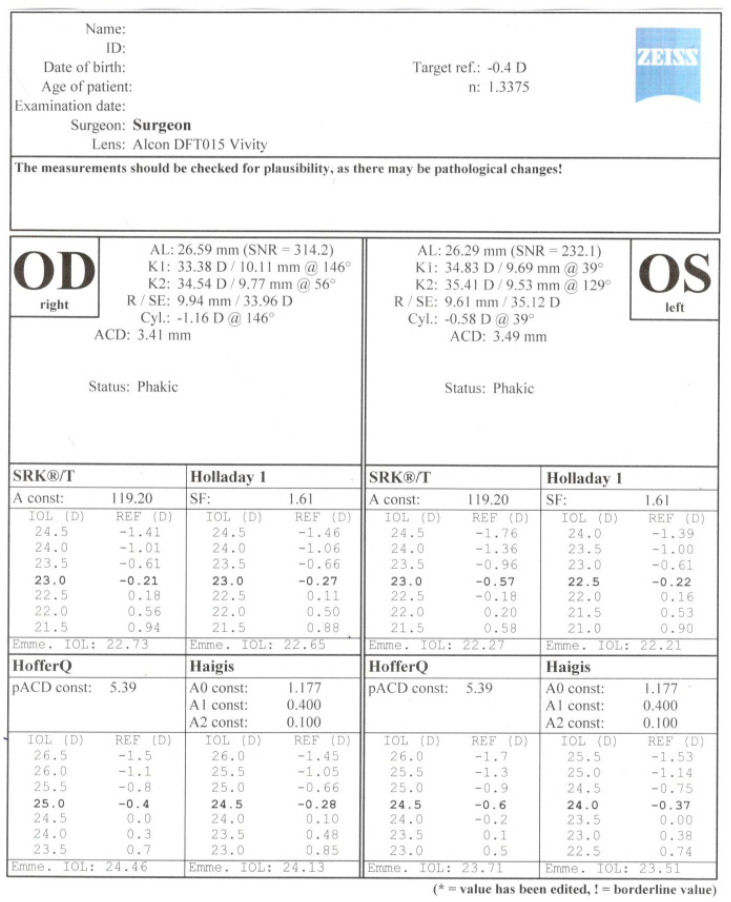
Result of intraocular lens power calculation using the IOL Master 500.

**Figure 3 medicina-58-00689-f003:**
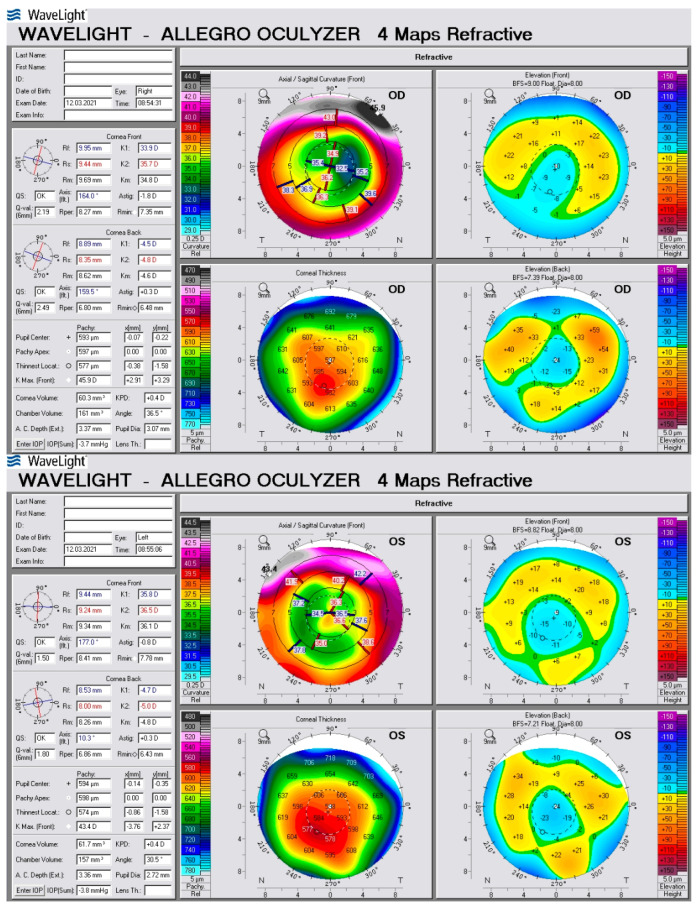
Corneal tomography shows irregular astigmatism in both eyes.

**Figure 4 medicina-58-00689-f004:**
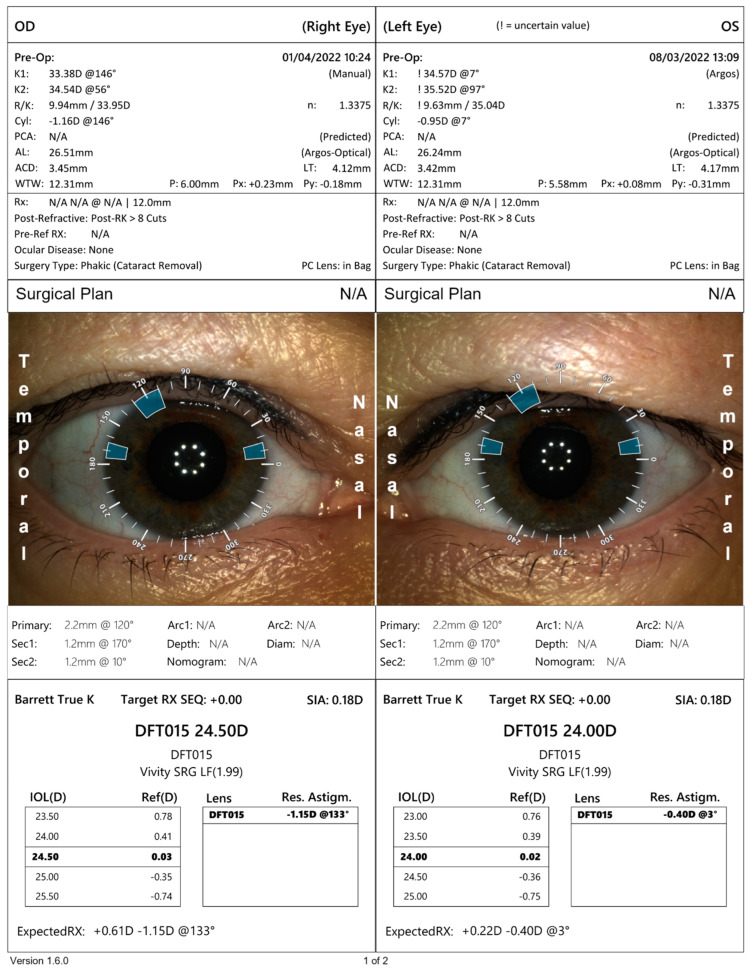
Result of intraocular lens power calculation using the Argos biometer.

**Figure 5 medicina-58-00689-f005:**
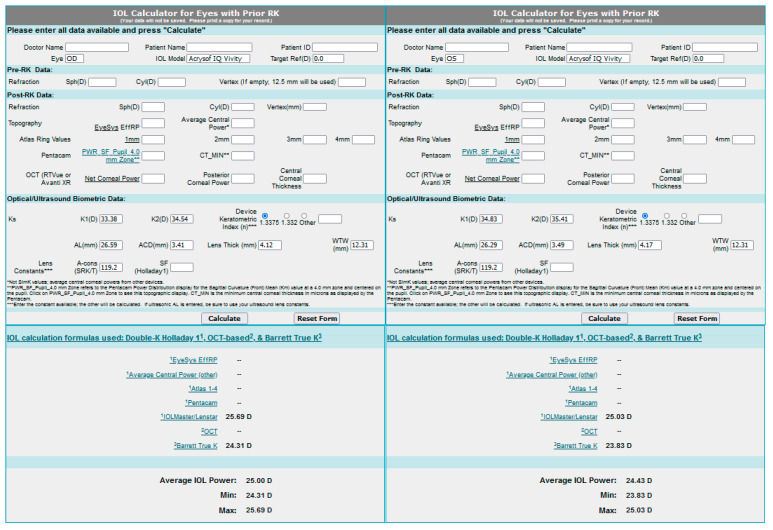
Result of intraocular lens power calculation using ASCRS calculator for the right and the left eye.

**Figure 6 medicina-58-00689-f006:**
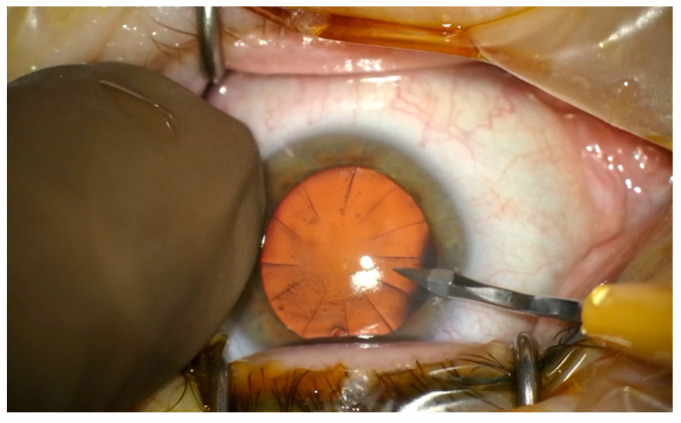
Location of lateral cuts between 2 neighboring incisions after radial keratotomy.

**Table 1 medicina-58-00689-t001:** Refraction and visual acuity (logMAR) before and after (2 weeks and 6 weeks) cataract surgery.

	Refraction	UCDVA	BCDVA	UCIVA	BCIVA	UCNVA	BCNVA
Preoperative data							
OD	+4.25/−0.75 ax 159	1.0	0.2	1.1	0.1	1.1	0.3
OS	+3.5/−0.5 ax 5	1.0	0.2	1.1	0.1	1.0	0.2
2 weeks after cataract surgery							
OD	+1.75/−1.75 ax 144	0.2	0.0	0.4	0.14	0.6	0.1
OS	−0.0/−0.5 ax 73	0.0	−0.1	0.5	0.1	0.5	0.1
6 weeks after cataract surgery							
OD	0.0/−1.25 ax 165	0.1	0.0	0.3	0.1	0.6	0.1
OS	−0.25/−0.25 ax 59	0.0	−0.1	0.0	0.0	0.4	0.0

Abbreviatons: OD—right eye, OS—left eye, UCDVA—uncorrected distance visual acuity at 4 m, BCDVA -best corrected distance visual acuity, UCIVA—uncorrected intermediate visual acuity at 80 cm, BCIVA—best corrected intermediate visual acuity, UCNVA—uncorrected near visual acuity at 40 cm, BCNVA—best corrected near visual acuity.

**Table 2 medicina-58-00689-t002:** Refraction and visual acuity (logMAR) 6 month after cataract surgery.

	Monocular		Binocular
	OD	OS	OU
Refraction	0.0/−1.25 ax 165	−0.5/−0.25 ax 59	
UCDVA	0.1	0.0	0.0
BCDVA	0.0 (0.0/−0.75 ax 165)	−0.1 (−0.25 Dsph)	−0.1
UCIVA	0.3	0.0	0.0
BCIVA	0.1 (0.5/−0.75 ax 165)	0.0	0.0
UCNVA	0.6	0.4	0.3
BCNVA	0.1 (1.0/−0.75 ax165)	0.0 (0.75 Dsph)	0.0

Abbreviatons: Dsph—Diopter sphera, OD—right eye, OS—left eye, OU—both eyes, UCDVA—uncorrected distance visual acuity at 4m, BCDVA—best corrected distance visual acuity, UCIVA—uncorrected intermediate visual acuity at 80 cm, BCIVA—best corrected intermediate visual acuity, UCNVA—uncorrected near visual acuity at 40 cm, BCNVA—best corrected near visual acuity.

## Data Availability

Not applicable.

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
