# Peer review of "Cataract Surgery after Radial Keratotomy with Non-Diffractive Extended Depth of Focus Lens Implantation"

_medicina, 2022, doi:10.3390/medicina58050689_

Round 1

Reviewer 1 Report

This is a report of a case with the history of RK in the past and successful refractive outcome following phaco with implantation of EDOF IOL.

Although the topic is interesting, the manuscript is not very well written. For instance, the case report includes information that should be seen in the discussion part, e.g. lines 50-57 and 69-76.

Also, the discussion part needs to be organised, discussing the below topics separately:

  • Choice of the IOL
  • Choice of the IOL power calculation
  • Incision type
  • And the refractive outcome of phaco in post-RK eyes based on the published papers.

And finally, the manuscript would benefit from English language modification.

Author Response

Thank you very much for reviewing my manuscript.

I have improved the layout of the manuscript by moving the suggested information from the case report section to the discussion. 

I divided the discussion into the listed topics and provided postoperative results from a publication on a EDOF lens implantation in patients after radial keratotomy. 

The manuscript was again checked by a native speaker for language accuracy.

Reviewer 2 Report

Thank you for choosing me to review this case report.

It details an extended depth of focus iol implantation in a radial keratotomy (RK) patient. This is usually not recommended as the refractive in radial keratotomy may not be stable, and keratometry may not be accurate.

My comments

  1. In the abstract, please mention how many cuts were made in the RK
  2. Abstract line 10: "In this study, we present"… this is not a study, so please change
  3. The intro is clinically and beautifully written. It has a nice flow and encompasses the difficulties in surgery in this patient, primarily patient expectation!
  4. Case report, line 34- in which eye did she feel deterioration of vision? Did she have more cataract related-symptoms (glare, mist, cloudy…etc..)
  5. How would you explain that she did not use any glass while her Rx was +4.5 (she was 60, and she had uncorrected VA 6\60)- what was her occupation? Did she drive? Is a VA test required in Poland before approving a driving license? In Israel, it is a criterion at the age of 50
  6. What is UCIVA?- why won't you use the full terms along with the text? It will make life easier for the reader.
  7. Why do you think there was 16 RK on re and 12 RK on LE
  8. Did you test the retina with OCT?
  9. Line 51 -58“Difficulties in determining the curvature 51 of the cornea and calculating the appropriate IOL power were caused by corneal irregularities after RK”…. this is all related to the discussions, not the case.
  10. “This IOL provides an extended range of vision from a distance with 69 excellent intermediate and functional near vision. It is based on non-diffractive X-wave 70 technology, which modifies the wave front and produces one elongated focus without 71 splitting light. Thanks to these properties, the lens reduces the risk of dysphotopsia. It 72 does not deteriorate the contrast sensitivity and is less sensitive to decentration than multifocal lenses. This lens is built from Acrysof, a hydrophobic material, and contains UV 74 and blue light filters. It has -1.5 D defocus and negative asphericity of the anterior surface 75 (-0.2 μm) [3], which is particularly important in patients with positive corneal aberrations.---move to the discussion.”
  11. Line 84 – unsuccessful in measuring OP? a typo?
  12. Which K did you use, the pentcam or the IOLmaster?
  13. So you used both the barret true k formula and the ACRS formula?
  14. What was the a-constant that you used? 119.2? on IOLCON, they suggest 119.08
  15. Why did you choose the 24 lens- the prediction was +0.41 in barret true K, and the  ASCRS recommends a stronger lens, 24.5 for Plano?
  16. Why do you think there was much difference between 2 and 4 weeks? Almost +1.7 D?
  17. For the left eye, does the calculator suggest a stronger lens than +23? Please explain your choice.
  18. Figure 6 I would suggest holding the eye with a cotton tip, a spear cut flash or forceps. Do you think that gloves increase the risk of infections?
  19. Table 1, can you please add baseline and week 2? It could also be in a chart.
  20. At what distance did you measure intermediate, near and far visual acuity?
  21. Is the aim of the EDOF to provide near or intermediate vision?
  22. “Currently, the patient does not require 130 spectacles for distance or intermediate distance”.= not surprising- well, she did not wear glasses even when she was 6/60 with +4 RX.
  23. Tables 2 and 1 could be plotted as well- it might be easier for the threader to see the dynamic.
  24. Table 2- the refraction is myopic in both eyes even when the IOL power was planned for hyperopic! Any comments on this effect? Is it because of the inaccurate measurement?
  25. Can you use such IOLs in the public sector as well? Did the patient have to purchase this IOL?

Overall this case provides new information about options for treatment in RK cases. As it can enhance patients' satisfaction. I would be happy to review once the authors have corrected/answered my comments. Please add your complete response under my numbered comments to make the review more convenient.

Author Response

Thank you very much for reviewing my manuscript.

  1. In the abstract, please mention how many cuts were made in the RK 

Thank you for your comment, I have added the number of incisions.

  1. Abstract line 10: "In this study, we present"… this is not a study, so please change

Thank you, I changed it to “In this case report, we present

  1. The intro is clinically and beautifully written. It has a nice flow and encompasses the difficulties in surgery in this patient, primarily patient expectation!

Thank you for your kind words.

  1. Case report, line 34- in which eye did she feel deterioration of vision? Did she have more cataract related-symptoms (glare, mist, cloudy…etc..)

The patient reported deterioration of vision in both eyes, foggy vision, increased halo and glare.

  1. How would you explain that she did not use any glass while her Rx was +4.5 (she was 60, and she had uncorrected VA 6\60)- what was her occupation? Did she drive? Is a VA test required in Poland before approving a driving license? In Israel, it is a criterion at the age of 50)

The patient had been wearing progressive glasses since 2004, prior to cataract surgery. This was not written in the article. In the manuscript, I wrote that the patient did not wear glasses 15 years after RK (1989).  Her visual acuity was very poor without glasses. She avoided driving, even with glasses, especially at night, because of her poor vision. 

In Poland, there is no age criteria at which visual acuity should be checked in order to obtain a driver's license. Each driver is obliged to undergo a visual acuity test of each eye checked separately as well as a test of both. Visual acuity must not be less than 0.5 in correction. 

  1. What is UCIVA?- why won't you use the full terms along with the text? It will make life easier for the reader.

The abbreviation UCIVA was explained when first used, in the case report. 

UCIVA means, uncorrected intermediate visual aquity, and it was examined from a distance of 80 cm.

  1. Why do you think there was 16 RK on re and 12 RK on LE

Due to the lack of medical documentation from the radial keratotomy procedure, I can only guess that the right eye initially had a higher refractive error. The patient reported that the glasses she wore before the radial keratotomy procedure did not fully correct her visual impairment, and the right eye was weaker.

  1. Did you test the retina with OCT?

Yes, the patient had OCT performed. OCT of the macula and OCT of the RNFL were normal. In the case report, I described that OCT was performed.

  1. Line 51 -58“Difficulties in determining the curvature 51 of the cornea and calculating the appropriate IOL power were caused by corneal irregularities after RK”…. this is all related to the discussions, not the case.

Thank you, I will move this part to the discussion.

  1. “This IOL provides an extended range of vision from a distance with 69 excellent intermediate and functional near vision. It is based on non-diffractive X-wave 70 technology, which modifies the wave front and produces one elongated focus without 71 splitting light. Thanks to these properties, the lens reduces the risk of dysphotopsia. It 72 does not deteriorate the contrast sensitivity and is less sensitive to decentration than multifocal lenses. This lens is built from Acrysof, a hydrophobic material, and contains UV 74 and blue light filters. It has -1.5 D defocus and negative asphericity of the anterior surface 75 (-0.2 μm) [3], which is particularly important in patients with positive corneal aberrations.---move to the discussion.”

Thank you, I will move this part to the discussion.

  1. Line 84 – unsuccessful in measuring OP? a typo?

Yes, that's a typo. I corrected it. It was the abbreviation OD.

  1. Which K did you use, the pentcam or the IOLmaster?

The IOL Master. 

  1. So you used both the barret true k formula and the ACRS formula?

Yes, we used both Barret True K formulas, the Argos biometer and the ACRS online calculator to select implant power.

  1. What was the a-constant that you used? 119.2? on IOLCON, they suggest 119.08

Then, when we counted the implant power for the patient (in September 2021) we used a constant A of 119.2. Now we use 119.08.

  1. Why did you choose the 24 lens- the prediction was +0.41 in barret true K, and the  ASCRS recommends a stronger lens, 24.5 for Plano?

In our experience, patients with longer axial length tend to have a myopic shift. Additionally, this was not the first patient after RK surgery in which we implanted the EDOF Vivity lens. An earlier patient also had a myopic shift, even though we planned to target emmetropia. Hence, the choice of a 24.0D lens in OD, and a 23.5 D lens in OS.

  1. Why do you think there was much difference between 2 and 4 weeks? Almost +1.7 D?

I think this is due to the peripheral corneal swelling between the post-RK incisions, also reported in the literature, which causes a transient hyperopic shift after cataract surgery.

  1. For the left eye, does the calculator suggest a stronger lens than +23? Please explain your choice.

Look at No. 15.

  1. Figure 6 I would suggest holding the eye with a cotton tip, a spear cut flash or forceps. Do you think that gloves increase the risk of infections?

The operator chose to slightly compress the eyeball with his own finger, as this facilitated the placement of the paracentesis between the two incisions after radial keratotomy. I think gloves, as well as other equipment equally can increase the risk of infection.

  1. Table 1, can you please add baseline and week 2? It could also be in a chart.

Of course, I will add the baseline data and 2 weeks after.

  1. At what distance did you measure intermediate, near and far visual acuity?

A standardized ETDRS chart at 4 m, 80 cm and 40 cm was used to measure VA.

  1. Is the aim of the EDOF to provide near or intermediate vision?

The EDOF Vivity lens offers good distance and intermediate vision.

  1. “Currently, the patient does not require 130 spectacles for distance or intermediate distance”.= not surprising- well, she did not wear glasses even when she was 6/60 with +4 RX.

My mistake, the patient wore glasses prior to cataract surgery.

  1. Tables 2 and 1 could be plotted as well- it might be easier for the threader to see the dynamic.

I have no idea how to visualize these results on a graph. I think that 2 tables with all the data (preoperative and postoperative) are a good way to show the changes in the patient's refraction and visual acuity.

  1. Table 2- the refraction is myopic in both eyes even when the IOL power was planned for hyperopic! Any comments on this effect? Is it because of the inaccurate measurement?

I think this is due to the difficulty of calculating implant power for patients after radial keratotomy and correctly calculating central corneal power. The patient has AL >26 mm, a deep anterior chamber, and in our experience, such patients have a myopic shift despite the target for emmetropia.

  1. Can you use such IOLs in the public sector as well? Did the patient have to purchase this IOL?

EDOF lenses in Poland are not implanted using public funds, like multifocal lenses. Only in appropriate cases can this lens be implanted in the public sector.